

# Substrate type and palaeodepth do not affect the Middle Jurassic taxonomic diversity of crinoids

Mariusz A. Salamon[1], Anna Feldman-Olszewska[2], Sreepat Jain[3], Bruno B.M. Ferré[4], Karolina Paszcza[5] and Bartosz J. Płachno[6]

[1] Faculty of Natural Science, University of Silesia in Katowice, Sosnowiec, Poland
[2] Polish Geological Institute - National Research Institute, Warszawa, Poland
[3] Department of Applied Geology, School of Applied Natural Sciences, Adama Science and Technology University, Adama, Ethiopia
[4] Saint Étienne du Rouvray, France
[5] Faculty of Natural Sciences, University of Silesie in Katowice, Sosnowiec, Poland
[6] Faculty of Biology, Institute of Botany, Jagiellonian University in Kraków, Kraków, Poland

Corresponding author
Mariusz A. Salamon,
paleo.crinoids@poczta.fm

## ABSTRACT

Crinoids are largely considered as good indicators for determining environmental conditions. They are robust proxies for inferring changes in salinity and sedimentation rate and for inferring substrate type. Some crinoid groups (*e.g.*, certain comatulids, cyrtocrinids, millericrinids) have a depth preference, thus, making them useful for palaeodepth estimation. The hypotheses that crinoid distribution is substrate-dependent (rock type) or palaeodepth-dependent is tested here based on (a) archival Bathonian-Callovian (Middle Jurassic) crinoid occurrences from Poland and (b) newer finds from five boreholes from eastern Poland. Qualitative data suggests that isocrinids and cyclocrinids occur in both carbonate and siliciclastic rocks. The cyrtocrinids and roveacrinids occur within carbonate rocks, whereas the comatulids are exclusive to siliciclastics. In terms of palaeodepth, most crinoid groups dominate in shallow environments with the sole exception of cyrtocrinids, that are ubiquitous and occur in both shallow (near shore and shallow marine) and slightly deeper (deeper sublittoral to open shelf) settings. The occurrences of the cosmopolitan taxa, Chariocrinus andreae and Balanocrinus subteres (isocrinids), is independent of both substrate type and palaeodepth. Quantitative analyses (Analysis Of Variance; ANOVA) based on substrate type, *i.e.*, substrate-dependency (claystones, sandstones and limestones), and palaeodepth *i.e.*, palaeodepth-dependency (near shore, shallow-marine, mid-ramp and offshore), corroborate qualitative results. Statistical analysis suggest that the distribution of crinoids shows a strong substrate-dependency but not for palaeodepth, although very weak significance (low p value) is noted for near shore and shallow marine settings and crinoid distribution.

## INTRODUCTION

The Middle Jurassic (especially Bajocian and Callovian) strata of the epicratonic Poland (Central European province = Submediterranean province) are well-known for their

excellently preserved and strongly diversified fossil fauna and flora (*e.g.*, *Kopik, 1974*; *Kopik, 1976*; *Kopik, 1979a*; *Kopik, 1979b*; *Kopik, 1998*; *Kopik, 2006*; *Zatoń, Marynowski & Bzowska, 2006a*; *Zatoń et al., 2006b*). These occur within clays and carbonate concretions and are represented by ammonites, gastropods, scaphopods, bivalves, belemnites, brachiopods, bryozoans, echinoids, asteroids, ophiuroids, wood fragments, and many others (for details see *Matyja & Wierzbowski, 2000*; *Matyja & Wierzbowski, 2003*; *Gedl et al., 2003*; *Kaim, 2004*; *Zatoń & Marynowski, 2004*; *Zatoń & Marynowski, 2006*; *Zatoń, Marynowski & Bzowska, 2006a*; *Zatoń et al., 2006b*; *Zatoń, 2010a*; *Zatoń, 2010b*; *Zatoń, Wilson & Zavar, 2011*). Crinoids, the group under study, are also diverse and abundant (more so in the boreholes of eastern Poland), and often form encrinites (a grain-supported bioclastic sedimentary rock) in which all or most of the grains are crinoid ossicles (*e.g.*, *Salamon & Feldman-Olszewska, 2018*). Despite such rich diversity and abundance, the crinoid data from the Middle Jurassic of Poland has largely been relegated to either brief mentions of species occurrences or of citing crinoid-yielding localities (see *Salamon & Feldman-Olszewska, 2018* and literature cited therein).

The present Middle Jurassic study, based on new data from five eastern Poland boreholes (see Fig. 1 and Table 1) and previously published crinoid records, has twin objectives: (a) to standardize crinoid taxonomic disparity with added remarks on systematic issues and previous collections, and (b) to test two hypotheses using this standardized dataset. These include: (1) Do crinoids occur equally and frequently in both terrigenous and carbonate facies (test for substrate-dependency)? (2) Do crinoids occur equally and frequently in both deep and shallow water environments (test for palaeodepth-dependency)? Besides, qualitative analysis, the analysis of variance (ANOVA) is used to quantitatively test these two hypotheses.

During the Bathonian-Callovian duration, the Polish Basin was part of the epicratonic Central European Basin that separated the Fennoscandian plains on the north from the Western Tethys to the south. Only the extreme southern part of Poland (Tatra Mountains and Pieniny Klippen Belt) was part of the Tethyan realm (see Fig. 1). These two realms display distinct provincial characteristics (as evident from the distribution of ammonites; see *Hallam, 1975*; *Brand, 1986*), but this provincialism is not noted for crinoids. Hence, this study, also analyses crinoid taxonomic diversity trends between these two palaeobiogeographic provinces, *i.e.,* Tethyan and Central European.

## Previous records

The Middle Jurassic (Callovian) crinoids of the epicratonic basins of Poland have been recorded from the Polish Jura Chain ('A' on Fig. 1) and the Mesozoic margin of the Holy Cross Mountains ('B' on Fig. 1). Early records only mentioned crinoid localities (*e.g.*, *Wójcik, 1910*; *Makowski, 1952*; *Różycki, 1953*). Later, *Dayczak-Calikowska (1980)* listed several taxa but did not mention the exact locations from where the crinoid specimens were collected or where they were kept (sample repository). Furthermore, neither description, nor illustration, or even basic taxonomic group assignment was provided; most, however, belong to balanocrinids. *Radwańska & Radwański (2003)* from northern Poland recorded Callovian columnals of *Cyclocrinus macrocephalus* and Oxfordian

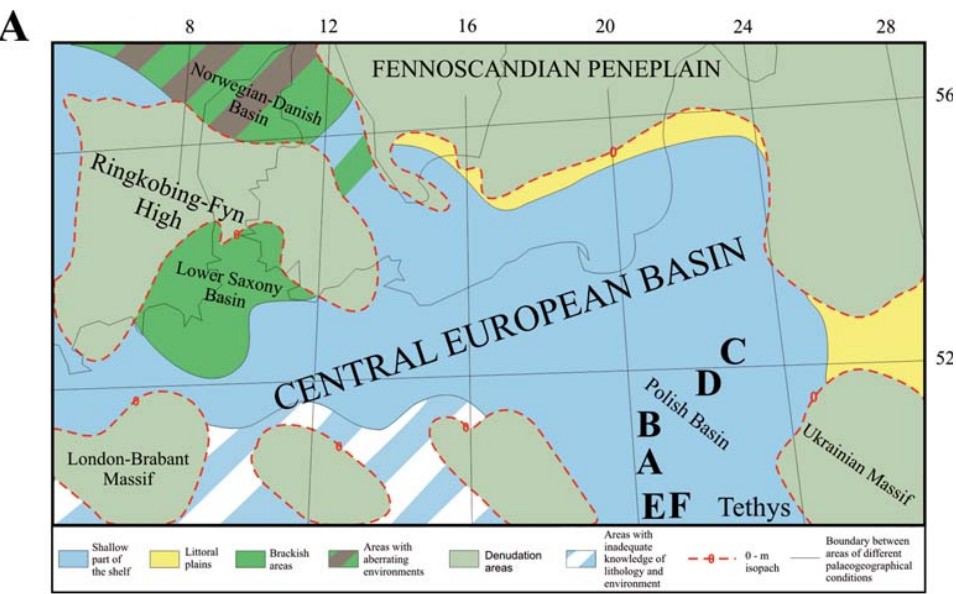

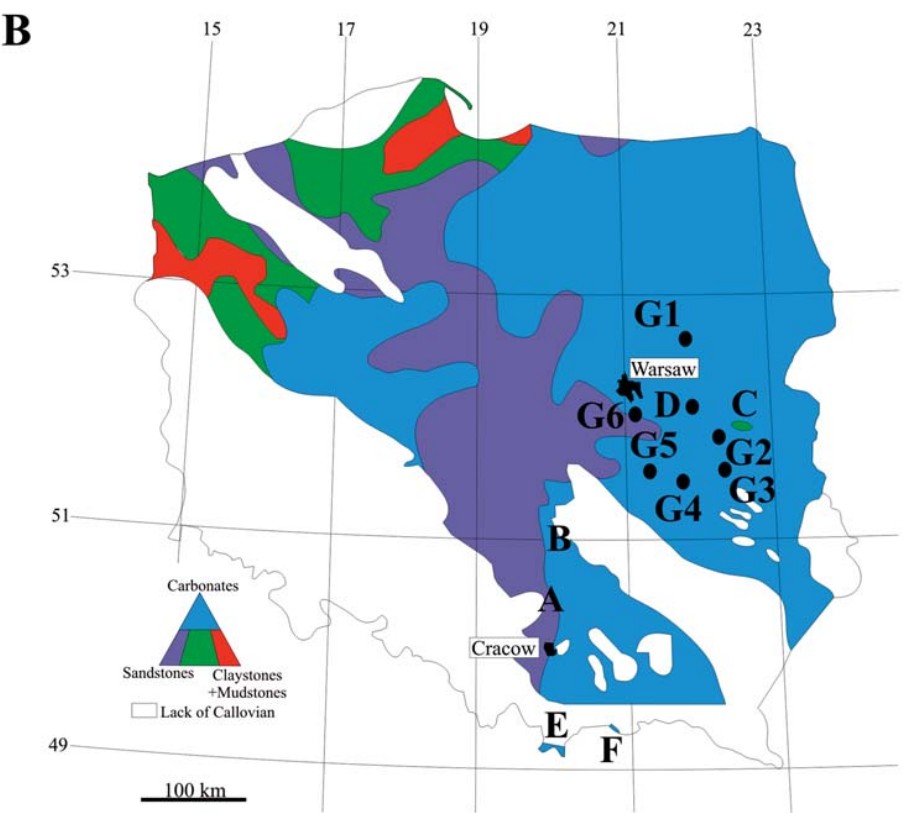

**Figure 1  Location maps.** (A) Palaeogeographic map of central Europe during the Middle Jurassic (modified after *Šimkevičius, Ahlberg & Grigelis, 2003*). (A) Bathonian and Callovian exposures of the southern part of the Polish Jura; (B) Bathonian and Callovian exposures of the southern margin of the Holy Cross Mountains; (continued on next page...)

**Figure 1 (…continued)**
(C) Callovian exposure of the glacial drift in Łuków; (G1-G6) location of the Bathonian-Callovian bore-holes described in this constribution [Tłuszcz IG1 (G1), Siedliska IG1 (G2), Kock IG2 (G3), Żyrzyn IG 1 (G4), Maciejowice IG1 (G5), Magnuszew IG1 (G6)], (D) location of borehole Żebrak IG1 (details in *Salamon & Feldman-Olszewska, 2018*); (A–D) Central European province; (E) Bathonian and Callovian exposures of the Tatra Mountains; (F) Bathonian and Callovian exposures of the Pieniny Klippen Belt; (E, F) Tethyan province. (B) Map of Callovian lithofacies noted in Poland (modified after *Dayczak-Calikowska & Moryc, 1988*).

cyclocrinid of *Cyclocrinus couiavianus*. *Salamon & Zatoń (2006)* from the Callovian marly limestones of the Zalas Quarry in southern Poland ('A' in Fig. 1; see also Table 2) erected a new species of balanocrinid (*Balanocrinus hessi*); this was later synonymized with *B. pentagonalis* by *Krajewski, Olchowy & Salamon (2019)*. *Salamon & Gorzelak (2007)*, from the same locality (Zalas Quarry in southern Poland; 'A' in Fig. 1) described few isolated remains of indeterminable cyrtocrinids associated with the cyrtocrinid cup of *Dolichocrinus* cf. *aberrans* (see Table 2). *Salamon & Zatoń (2007)* from a collection of ~1,500 remains of columnals, pluricolumnals, brachials and cups, illustrated 11 crinoid taxa, including some isocrinids, comatulids, millecrinids and cyclocrinids (see Table 2). These specimens came from three Bajocian, four Bathonian and two Callovian localities of the southern part of the Polish Jura Chain and the Mesozoic margin of the Holy Cross Mountains (see *Salamon & Zatoń, 2007*). Current re-examination of this material led to the assignment of the columnals described as *Millericrinina* to the cyrtocrinids, Cyrtocrinida indet. (see *Salamon & Feldman-Olszewska, 2018*) (Table 2). *Salamon (2008a)* and *Salamon (2008b)* from two Callovian localities (Polish Jura Chain, 'A' in Fig. 1 and the "glacial drift" of Łuków in eastern Poland: 'C' in Fig. 1) recorded several crinoid taxa. The "glacial drift" assemblage of Łuków ('C' in Fig. 1) is dominated by isocrinids with associated comatulids. The carbonate rocks of the Polish Jura Chain yielded a rich assemblage of cyrtocrinids with dozens of complete individuals and a few isolated isocrinid remains. *Salamon & Feldman-Olszewska (2018)* from the Callovian crinoidal limestones of the Żebrak IG 1 borehole (eastern Poland; 'D' in Fig. 1) described a collection of five isocrinid taxa with some unidentifiable cyrtocrinids (see also Table 2). This collection contained a relatively small number of complete or nearly complete individuals, as a side-effect of the maceration process. Before the maceration process, the samples formed a typical encrinite and consisted largely of crinoids, as is also the case in the present study, yielding only fragmentary crinoids (see Table 2). Additionally, the Middle Jurassic (Bathonian-Callovian carbonates) exposures in southern Poland (Tethyan province, 'E' and 'F' in Fig. 1) have also yielded isocrinids, cyrtocrinids, roveacrinids, and cyclocrinids (see *Głuchowski, 1987*; see Table 2).

## MATERIAL AND METHODS

The archival cores (72 core samples) drilled in eastern Poland (G1–G6 on Figs. 1 and 2) were investigated for crinoids. These cores are stored in the Polish Geological Institute National Research Institute, Warsaw (Poland). The samples selected for the maceration process were selected from the following boreholes at respective depths: (1) Kock IG 2 (core depth: 859.0 m; age: Callovian?); (2) Maciejowice IG 1 (core depth: 1,431.5 m; age: late Bathonian);

**Table 1** Bathonian-Callovian crinoids collected in particular boreholes in eastern Poland.

| Borehole | Depth (in m) | Age | Lithology | Determinable crinoid taxa |
|---|---|---|---|---|
| Kock IG 2 | 859.0 | Callovian? | Yellowish grey, crinoidal limestone, packstone composed of echinoderm plates, bivalve shells, rare bryozoa fragments with ferruginous impregnations, ferruginous ooids, rounded quartz grains and sparite cement | Isocrinida indet. *Balanocrinus subteres* |
| Maciejowice IG 1 | 1,431.5 | late Bathonian | Dark grey, medium-size sandstone | none |
| Magnuszew IG 1 | 1,505.7 | late Callovian | Yellowish grey, crinoidal limestone, grainstone composed of echinoderm plates, echinoid spines, rounded quartz grains, sparite cement and iron oxides/hydroxides | Isocrinida indet. *Balanocrinus subteres* *Pentacrinites dargniesi* |
| | 1,507.3 | late Callovian | Yellowish grey, crinoidal limestone with amonite and belemnite, grainstone composed of echinoderm plates, echinoid spines, bivalve shells, rounded quartz grains, sparite cement and iron oxides/hydroxides | Isocrinida indet. *Pentacrinites dargniesi* |
| | 1,508.65 | late Callovian | Grey crinoidal limestone, grainstone composed of echinoderm plates, echinoid spines, bivalve shells, bryozoa fragments, rare gastropoda, rounded quartz grains and sparite cement | Isocrinida indet. *Balanocrinus subteres* *Pentacrinites dargniesi* |
| | 1,522.4 | middle-late Callovian | Yellow-brown, dolomitic crinoidal limestone with foraminifera, quartz grains and iron oxides/hydroxides | Isocrinida indet. *Isocrinus nicoleti* *Balanocrinus subteres* *Balanocrinus pentagonalis* *Pentacrinites dargniesi* |
| Siedliska IG 1 | 824.65 | early Callovian | Beige-grey, crinoidal limestone, grainstone composed of echinoderm plates, echinoid spines and bryozoa fragments frequently with ferruginous impregnations, crushed bivalve and brachiopods shells, gastropods, lithoclasts with iron hydroxides, rare ferruginous ooids, quartz grains, sparite or sparite-micrite cement and iron oxides/hydroxides | Isocrinida indet. *Pentacrinites dargniesi* |
| | 828.25 | early Callovian | Beige-grey, crinoidal limestonegrainstone composed of echinoderm plates, echinoid spines and bryozoa fragments frequently with ferruginous impregnations, crushed bivalve and brachiopods shells, gastropods, lithoclasts with iron hydroxides, rare ferruginous ooids, quartz grains, sparite or sparite-micrite cement and iron oxides/hydroxides | Isocrinida indet. *Balanocrinus subteres* *Pentacrinites dargniesi* |
| | 828.77 | early Callovian | Beige-grey, crinoidal limestone grainstone composed of echinoderm plates, echinoid spines and bryozoa fragments frequently with ferruginous impregnations, crushed bivalve and brachiopods shells, gastropods, lithoclasts with iron hydroxides, rare ferruginous ooids, quartz grains, sparite or sparite-micrite cement and iron oxides/hydroxides | Isocrinida indet. *Isocrinus nicoleti* *Balanocrinus subteres* *Pentacrinites dargniesi* Cyrtocrinida indet. |
| | 830.8 | late Bathonian? | Yellow-brown, crinoidal limestone with abundant iron hydroxides, lithoclasts and bivalve shells | none |
| | 837.65 | late Bathonian? | Yellow crinoidal limestone, grainstone composed of echinoderm plates, echinoid spines and bryozoa fragments frequently with ferruginous impregnations, crushed bivalve and brachiopods shells, gastropods, lithoclasts with iron hydroxides, rare ferruginous ooids, quartz grains, sparite or sparite-micrite cement and iron oxides/hydroxides | Isocrinida indet. *Chariocrinus andreae* *Balanocrinus subteres* *Balanocrinus pentagonalis* *Pentacrinites dargniesi* |

**Table 1** (*continued*)

| Borehole | Depth (in m) | Age | Lithology | Determinable crinoid taxa |
|---|---|---|---|---|
| Tłuszcz IG 1 | 1,047.7 | late Callovian?-early Oxfordian | Beige-grey, crinoidal limestone | none |
| | 1,049.95 | late Callovian | Cream crinoidal limestone, grainstone composed of echinoderm plates, echinoid spines and bryozoa fragments, bivalve shells, very rare foraminifera, sparite cement and iron oxides/hydroxides | Isocrinida indet. *Balanocrinus pentagonalis* *Pentacrinites dargniesi* *Phyllocrinus* sp. |
| Żyrzyn IG 1 | 1,134.95 | Callovian | Yellowish grey, crinoidal limestone, grainstone composed of echinoderm plates, bryozoa fragments, bivalve shells, quartz grains, sparite cement and iron oxides/hydroxides | Isocrinida indet. *Chariocrinus andreae* *Pentacrinites dargniesi* Cyrtocrinida indet. |
| | 1,137.7 | late Bathonian? | Beige-grey, crinoidal limestone with cement and iron hydroxides | none |
| | 1,147.9 | late Bathonian | Beige-grey, crinoidal limestone, grainstone composed of echinoderm plates, echinoid spines, bryozoa fragments, bivalve and brachiopods shells, quartz grains, sparite cement and iron oxides/hydroxides | Isocrinida indet. *Isocrinus nicoleti* *Chariocrinus andreae* *Balanocrinus subteres* *Pentacrinites dargniesi* Cyrtocrinida indet. |
| | 1,155.4 | late Bathonian | Yellow-brown, crinoidal limestone, grainstone composed of echinoderm plates, bryozoa fragments, bivalve and brachiopods shells, sparite cement and iron oxides/hydroxides | Isocrinida indet. *Isocrinus nicoleti* *Balanocrinus pentagonalis* *Pentacrinites dargniesi* |

**Table 2  Bathonian and Callovian crinoid list recorded in different areas of Poland.**

| Location area (see also details in Fig. 1 and its caption) | Age | Number of crinoid taxa | List of crinoids |
|---|---|---|---|
| C –glacial drift in Łuków; Central European province | Callovian | 10, including: 8 isocrinids and 2 comatulids | **Isocrinids:** *Chariocrinus andreae* *Balanocrinus berchteni* *Balanocrinus pentagonalis* *Balanocrinus subteres* *Isocrinus* sp. *Isocrinus nicoleti* *Isocrinus pendulus* *Pentacrinites* cf. *dargniesi* **Comatulids:** Paracomatulidae sp. et gen. indet. *Palaeocomaster* sp. |
| G1-G6 –boreholes in eastern Poland (Kock IG 2, Maciejowice IG 1, Magnuszew IG 1, Siedliska IG 1, Tłuszcz IG 1, Żyrzyn IG 1); Central European province | latest Bathonian-Callovian | 8, including: 6 isocrinids and 2 cyrtocrinids | **Isocrinids:** Isocrinida indet. *Chariocrinus andreae* *Balanocrinus subteres* *Balanocrinus pentagonalis* *Isocrinus nicoleti* *Pentacrinites dargniesi* **Cyrtocrinids:** Cyrtocrinida indet. *Phyllocrinus* sp. |
| D –Żebrak IG 1; Central European province | Callovian | 6, including: 5 isocrinids and 1 cyrtocrinid | **Isocrinids:** *Chariocrinus andreae* *Balanocrinus* cf. *subteres* *Isocrinus* sp. *Isocrinus* cf. *nicoleti* *Pentacrinites* sp. **Cyrtocrinids:** Cyrtocrinida indet. |
| B –southern margin of the Holy Cross Mountains; Central European province | Bathonian | 3, including: 2 isocrinids and 1 comatulid | **Isocrinids:** *Chariocrinus andreae* *Balanocrinus berchteni* **Comatulids:** *Paracomatula helvetica* |
| B –southern margin of the Holy Cross Mountains; Central European province | Callovian | 2, including: 1 isocrinid and 1 cyclocrinid | **Isocrinids:** *Chariocrinus andreae* **Cyclocrinids** (order uncertain according to *Hess & Messing (2011)*): *Cyclocrinus macrocephalus* |

**Table 2** (*continued*)

| Location area (see also details in **Fig. 1** and its caption) | Age | Number of crinoid taxa | List of crinoids |
|---|---|---|---|
| A –southern part of the Polish Jura; Central European province | Bathonian | 6, including: 3 isocrinids, 2 comatulids and 1 cyrtocrinid | **Isocrinids:** *Chariocrinus andreae* *Balanocrinus berchteni* *Isocrinus bajociensis* **Comatulids:** *Paracomatula helvetica* *Palaeocomaster* sp. **Cyrtocrinids:** Cyrtocrinida indet. (=Millericrinina in *Salamon & Zatoń (2007)*) |
| A –southern part of the Polish Jura; Central European province | Callovian | 15, including: 3 isocrinids, 11 cyrtocrinids and 1 cyclocrinid | **Isocrinids:** *Chariocrinus andreae* *Balanocrinus subteres* *Balanocrinus pentagonalis* (=*Balanocrinus hessi* in *Salamon & Zatoń (2006)*) **Cyrtocrinids:** Cyrtocrinida indet. (=Millericrinina in *Salamon & Zatoń (2007)*) *Phyllocrinus* sp. *Phyllocrinus stellaris* *Phyllocrinus belbekensis* *Tetracrinus moniliformis* *Sclerocrinus* sp. *Cyrtocrinus* sp. *Pilocrinus moussoni* *Dolichocrinus aberrans* *Fischericrinus ausichi* *Lonchocrinus dumortieri* **Cyclocrinids:** *Cyclocrinus macrocephalus* |
| E –Tatra Mountains; Tethyan province | Bathonian | 8, including: 6 isocrinids, 1 cyrtocrinid and 1 cyclocrinid | **Isocrinids:** *Isocrinus* sp. *Isocrinus bathonicus* *Isocrinus nicoleti* *Isocrinus pendulus* *Balanocrinus* sp. *Pentacrinites dargniesi* **Cyrtocrinids:** Cyrtocrinida indet. **Cyclocrinids:** *Cyclocrinus rugosus* |
| E –Tatra Mountains; Tethyan province | Callovian | 3, including: 1 isocrinid, 1 cyrtocrinid and 1 roveacrinid | **Isocrinids:** *Isocrinus* sp. **Cyrtocrinids:** Cyrtocrinida indet. **Roveacrinids:** *Saccocoma* sp. |

**Table 2** (*continued*)

| Location area (see also details in Fig. 1 and its caption) | Age | Number of crinoid taxa | List of crinoids |
|---|---|---|---|
| F –Pieniny Klippen Belt; Tethyan province | Bathonian | 16, including: 10 isocrinids, 5 cyrtocrinids and 1 roveacrinid | **Isocrinids:** *Chariocrinus andreae* *Isocrinus* sp. *Isocrinus bathonicus* *Isocrinus nicoleti* *Isocrinus pendulus* *Balanocrinus* sp. *Balanocrinus berchteni* *Balanocrinus pentagonalis* *Balanocrinus subteres* *Pentacrinites dargniesi* **Cyrtocrinids:** Cyrtocrinida indet. *Plicatocrinus tetragonus* *Lonchocrinus dumortieri* *Remisovicrinus* sp. *Dolichocrinus* sp. *aberrans* **Roveacrinids:** *Saccocoma* sp. |
| F –Pieniny Klippen Belt; Tethyan province | Callovian | 14, including: 2 isocrinids, 9 cyrtocrinids, 1 roveacrinid, and 2 cyclocrinids | **Isocrinids:** *Balanocrinus subteres* *Balanocrinus pentagonalis* **Cyrtocrinids:** Cyrtocrinida indet. *Lonchocrinus dumortieri* *Remisovicrinus* sp. *Dolichocrinus* cf. *aberrans* *Eudesicrinus* sp. *Pilocrinus moussoni* *Gammarocrinites* sp. *Sclerocrinus compressus* *Phyllocrinus* sp. **Roveacrinids:** *Saccocoma* sp. **Cyclocrinids:** *Cyclocrinus* sp. *Cyclocrinus rugosus* |

(3) Magnuszew IG 1 (core depths: 1,505.7 m, 1,507.3 m, 1,508.65 m, and 1,522.4 m; age: middle and late Callovian respectively); (4) Siedliska IG 1 (core depths: 824.65 m, 828.25 m, 828.77 m, 830.8 m, and 837.65 m; age: late Bathonian?-early Callovian); (5) Tłuszcz IG 1 (core depth: 1,047.7 m and 1,049.95 m; age: late Callovian?-early Oxfordian); (6) Żyrzyn IG 1 (core depths: 1,134.95 m, 1,137.7 m, 1,147.9 m, and 1,155.4 m; age: late Bathonian-Callovian) (for a summary see Table 2).

The present paper also includes a large collection of Callovian crinoids from Poland collected by Mariusz Salamon and Bartosz Płachno (2006-present) and others, mentioned in the *Acknowledgements* section.

The first step for the current analysis consisted of examining the drill cores under a binocular microscope for crinoids. Thereafter, the carbonate samples were soaked with

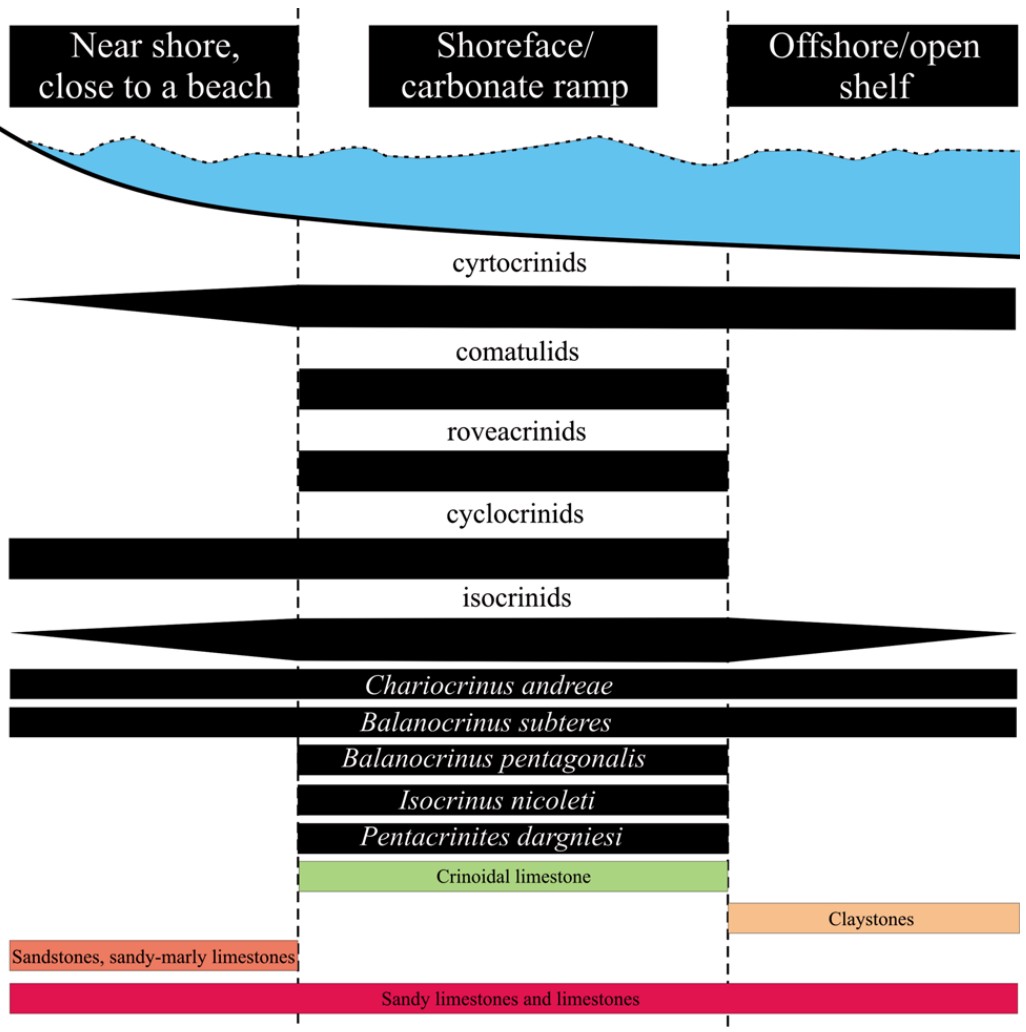

**Figure 2** **Model showing distribution of selected Callovian crinoids.**

Glauber's salt. Based on sample cohesion, these were then successively boiled and frozen. The residue was then washed under running tap water and sieved (mesh sizes: Ø1.0, 0.315 and 0.1 mm, respectively). The final step consisted of drying the screened residue at 170–180 °C (following the methodology outlined by *Krajewski, Olchowy & Salamon, 2019*).

Both the sandstone sample Maciejowice IG 1 (core depth: 1,431.5 m; age: late Bathonian) and the carbonate sample Magnuszew IG 1 (core depth: 1,505.7 m; age: late Callovian) were naturally macerated and left outside for 30 days (from January 22nd to February 21st, 2021). As a result of this natural maceration, the samples were partially disintegrated. In addition to this, several thin sections (TS) and polished slabs (PS) were also made. All crinoids were hand-picked from the maceration residue and photographed using a Canon Eos 350D digital camera, LeicaWildM10 coupled with a NikCamPro1 microscope and a Scanning Electron Microscope Philips XL-20. All specimens are housed in the Institute

of Earth Sciences of the University of Silesia in Katowice, Poland, and catalogued under registration number GIUS 8-3734. Other Middle Jurassic specimens used in the current study are also housed at the Institute of Earth Sciences of the University of Silesia in Katowice, Poland, and catalogued under: GIUS 8-2510, 8-2569, 8-2571, 8-3460Cr, 8-3466, 8-3678/1-6, 8-3734.

## Statistical methods

The taxonomically standardized dataset was subjected to Analysis Of Variance (ANOVA). To check for substrate-dependency (Hypothesis 1), the dataset was categorized under claystones, sandstones and limestones (the former two represent siliciclastics, whereas the latter, carbonates). For palaeodepth-dependency (Hypothesis 2), the dataset was categorized under near shore, shallow-marine, mid-ramp and offshore categories. For pairwise comparisons, the Tukey's HSD (honestly significant difference) test was applied.

## Taxonomic standardisation

The cyrtocrinids are identified at the specific level based on their cups. In some cases, when the cups are fragmentary or some typical features are not visible, the samples were assigned to the generic level, only. Most of the disarticulated remains of cyrtocrinids (holdfasts, columnals, radials, basals, brachials, etc.) are assigned under Cyrtocrinida indet.

As for comatulid centrodorsals, where cups with basals and radials were not available for additional information, they were also classified at the generic level. The brachial with muscular articulation on the proximal side and syzygial one on the distal side are classified as Paracomatulidae sp. et gen. indet.

The isolated remains of isocrinids, consisting of columnals, pluricolumnals, cirrals, cup plates, and brachials, are classified at the specific level. If they were found in different levels, they are described as Isocrinida indet.

A complete individual of cyclocrinid has not been found so far. Despite this, the uniqueness of their remains (*e.g.*, large cylindrical columnals with peculiar tuberculate facets) allows us to identify them at the specific level (see detailed discussion in *Radwańska & Radwański (2003)*. *Głuchowski (1987)* assigned some remains from the Pieniny Klippen Belt to *Cyclocrinus* sp. due to the incompleteness of columnals. *Głuchowski (1987)* classified all roveacrinids (saccocomids) as *Saccocoma* sp.; these were recorded either as isolated cup remains or observed in thin sections (TS).

Thus, the dataset used in the present study is taxonomically standardised. The crinoids classified in the present study are given in Tables 1–4. A summary of all data with their respective inferred palaeodepth and species diversity (number of taxa) is given in Table 3 (see also Fig. 3).

## RESULTS

The present study identified several crinoid remains of cups, columnals, cirrals and brachials from the 5 boreholes drilled in eastern Poland (Fig. 1). Only two pluricolumnals were recorded. The first one consists of three or four columnals observable on the surface of a rock fragment from the Magnuszew IG 1 core (depth: 1,505.7 m). The second one consists

**Table 3 Distribution of Bathonian-Callovian crinoids recorded in Poland within different facies and setting.**

| Lithology | Locality | Age | Energy level | Palaeodepth | Substrate | Isocrinida indet. | Chariocrinus andreae | Balanocrinus sp. | Balanocrinus berchteni | Balanocrinus pentagonalis | Balanocrinus subteres | Isocrinus sp. | Isocrinus nicoleti | Isocrinus pendulus | Isocrinus bajociensis | Isocrinus bathonicus | Pentacrinites dargniesi | Paracomatulidae sp. et gen. indet. | Palaeocomaster sp. | Paracomatula helvetica | Cyrtocrinida indet. | Phyllocrinus sp. | Phyllocrinus stellaris | Phyllocrinus belbekensis | Tetracrinus moniliformis | Sclerocrinus sp. | Sclerocrinus compressus | Cyrtocrinus sp. | Pilocrinus moussoni | Dolichocrinus aberrans | Fischericrinus ausichi | Lonchocrinus dumortieri | Plicatocrinus tetragonus | Remisovicrinus sp. | Eudesicrinus sp. | Gammarocrinite s sp. | Saccocoma sp. | Cyclocrinus sp. | Cyclocrinus macrocephalus | Cyclocrinus rugosus |
|---|---|---|---|---|---|---|---|---|---|---|---|---|---|---|---|---|---|---|---|---|---|---|---|---|---|---|---|---|---|---|---|---|---|---|---|---|---|---|---|---|
| **Abiotic factors** | | | | | | 1 | 5 | 2 | 4 | 2 | 2 | 2 | 2 | 2 | 3 | 1 | 2 | 3 | 3 | 3 | 2 | 1 | 1 | 1 | 1 | 1 | 1 | 1 | 1 | 2 | 1 | 1 | 1 | 1 | 1 | 1 | 1 | 1 | 4 | 1 |
| Claystones | B – southern margin of the Holy Cross Mountains | Bathonian | Low energy to moderately turbulent | Offshore below the wave base | | | 20 | | | 200 | | | | | | | | | | | 3 | | | | | | | | | | | | | | | | | | | | |
| Claystones with associated carbonate concretions | C –glacial drift at Łuków | Callovian | Moderately turbulent to turbulent | Shallow-marine | Siliciclastics | | 104 | 17 | 101 | 5 | 33 | 87 | 17 | 7 | | | 11 | 5 | 1 | | | | | | | | | | | | | | | | | | | | | | |
| Claystones with levels of siderites and carbonate concretions | A – southern part of the Polish Jura | Bathonian | Low energy to moderately turbulent | Offshore below the wave-base | | | 60 | | 800 | | | | | | 4 | | | | 1 | 5 | | | | | | | | | | | | | | | | | | | | |
| Organodetrital limy sandstones, sandy-marly limestones and marls | B – southern margin of the Holy Cross Mountains | Callovian | Turbulent | Near shore, close to a beach | | | 10 | | | | | 10 | | | | | | | | | | 4 | | | | | | | | | | | | | | | | | | 24 | |
| Sandstones | A – southern part of the Polish Jura | Callovian | Turbulent | Near shore, close to a beach | | | 30 | | | | | | | | | | | p | | | | | | | | | | | | | | | | 1 | | | | | | | |
| Sandy-limestones and limestones | A – southern part of the Polish Jura | Callovian | Low energy to moderately turbulent | Deeper sublittoral with influence of storms to open shelf | | | 35 | | | | | 45 | | | | | | | 1 | 13 | 4 | 6 | | 3 | | 3 | 8 | 3 | ca. 10 | 7 | | | | | | 6 | | | | |
| Crinoidal limestones | D – boreholes in eastern Poland (Kock IG 2, Maciejowice IG 1, Siedliska IG 1, Tłuszcz IG 1, Żyrzyn IG 1) | Uppermost Bathonian-Callovian | Turbulent | Shallow-marine | | | ca. 100 | ca. 100 | | | 28 | ca. 120 | | ca. 70 | | | | ca. 80 | | | | ca. 150 | 2 | | | | | | | | | | | | | | | | | |
| Crinoidal limestones | D –Żebrak IG 1 | Callovian | Turbulent | Shallow-marine | | | 40 | | | | | 4 | 30 | 9 | | | | 47 | | | | 10 | | | | | | | | | | | | | | | | | | |
| Crinoidal limestones | E –Tatra Mountains | Bathonian | Turbulent | Shallow-marine | | | | | p | | | | p | p | p | | p | p | | | | p | | | | | | | | | | | | | | | | | p | |
| Crinoidal limestones | E –Tatra Mountains | Callovian | Turbulent | Shallow-marine | | | | | | | | | p | | | | | p | | | | | | | | | | | | | | | | | | p | | | | |
| Crinoidal limestones | F –Pieniny Klippen Belt | Bathonian | Turbulent | Shallow-marine | Carbonate | | p | | p | p | p | p | p | p | p | | p | p | | | | p | | | | | | | p | p | p | | | p | | | | | | p | |
| Crinoidal limestones | F –Pieniny Klippen Belt | Callovian | Turbulent | Shallow-marine | | | | | | | p | p | | | | | | p | | | | | | p | | p | p | | p | | p | p | p | p | | | | | | p | |

*(continued on next page)*

**Table 3** (*continued*)

| Palaeodepth | Isocrinida indet. | Chariocrinus andreae | Balanocrinus sp. | Balanocrinus berchteni | Balanocrinus pentagonalis | Balanocrinus subteres | Isocrinus sp. | Isocrinus nicoleti | Isocrinus pendulus | Isocrinus bajociensis | Isocrinus bathonicus | Pentacrinites dargniesi | Paracomatulidae sp. et gen. indet. | Palaeocomaster sp. | Paracomatula helvetica | Cyrtocrinida indet. | Phyllocrinus sp. | Phyllocrinus stellaris | Phyllocrinus belbekensis | Tetracrinus moniliformis | Sclerocrinus sp. | Sclerocrinus compressus | Cyrtocrinus sp. | Pilocrinus moussoni | Dolichocrinus aberrans | Fischericrinus ausichi | Lonchocrinus dumortieri | Plicatocrinus tetragonus | Remisovicrinus sp. | Eudesicrinus sp. | Gammarocrinites sp. | Saccocoma sp. | Cyclocrinus sp. | Cyclocrinus macrocephalus | Cyclocrinus rugosus |
|---|---|---|---|---|---|---|---|---|---|---|---|---|---|---|---|---|---|---|---|---|---|---|---|---|---|---|---|---|---|---|---|---|---|---|---|
| 6 | | 20 | | 200 | | | | | | | | | | | | 3 | | | | | | | | | | | | | | | | | | | |
| 6 | | 104 | 17 | 101 | 5 | 33 | 87 | 17 | 7 | | | 11 | | 5 | 1 | | | | | | | | | | | | | | | | | | | | |
| 6 | | 60 | | 800 | | | | | | 4 | | | | 1 | 5 | | | | | | | | | | | | | | | | | | | | |
| 6 | | 10 | | | | 10 | | | | | | | | | | 4 | | | | | | | | | | | | | | | | | | 24 | |
| 6 | | 10 | | | | 10 | | | | | | | | | | 4 | | | | | | | | | | | | | | | | | | 24 | |
| 6 | | | | | | 30 | | | | | | | | | | | | | | | | | | | | | | | | | | | | | |
| 7 | | 35 | | | | 45 | | | | | | | | | | | 1 | 13 | 4 | 6 | 3 | | 3 | 8 | | | ca. 10 | 7 | | | | | | 6 | |
| 6 | ca. 100 | ca. 100 | | | 28 | ca. 120 | | ca. 70 | | | | ca. 80 | | | | ca. 150 | 2 | | | | | | | | | | | | | | | | | | |
| 6 | | 40 | | | | 4 | 30 | 9 | | | | 47 | | | | 10 | | | | | | | | | | | | | | | | | | | |
| 6 | | P | | | | | P | P | P | | P | P | | | | P | | | | | | | | | | | | | | | | | | | P |
| 6 | | | | | | | P | | | | | | | | | P | | | | | | | | | | | | | | | | | P | | |
| 6 | | P | P | P | P | P | P | P | P | | P | P | | | | P | | | | | | | | | | P | | P | P | P | | P | | | |
| 6 | | | | | P | P | | | | | | | | | | P | | | | | | P | | P | P | | P | | P | P | P | P | P | | P |
| 6 | | | | | P | P | | | | | | | | | | P | | | | | | P | | P | P | | P | | P | P | P | P | P | | P |
| 6 | | | | | | | | | | | | | | | | | | | | | | | | | | | | | | | | | | | |

**Notes.**

1, Exclusively in carbonates (column); 2, Predominance in carbonates (column); 3, Exclusively in siliciclastics (column); 4, Predominance in siliciclastics (column); 5, Cosmopolitan (column); 6, Near shore and shallow marine (row); 7, Deep sublittoral with influence of storms to open shelf (row).

**Table 4  Distribution of Bathonian-Callovian crinoids recorded in Poland within different facies and settings.**

| Species | Details |
|---|---|
| Isocrinida indet. | Plenty of more or less complete columnals, brachials, and cirrals |
| *Isocrinus nicoleti* (Desor, 1845) | Plenty of more or less complete, isolated columnals |
| *Chariocrinus andreae* (Desor, 1845) | Plenty of more or less complete, isolated columnals |
| *Balanocrinus subteres* (Münster in Goldfuss, 1826) | Plenty of more or less complete, isolated columnals |
| *Balanocrinus pentagonalis* (Goldfuss, 1826-1844) | 21 almost complete columnals |
| *Pentacrinites dargniesi* Terquem & Jourdy, 1869 | Plenty of cirrals, 14 almost complete columnals |
| Cyrtocrinida indet. | Plenty of cup remains and partly preserved columnals and 27 nearly complete columnals |
| *Phyllocrinus* sp. | Two cups |

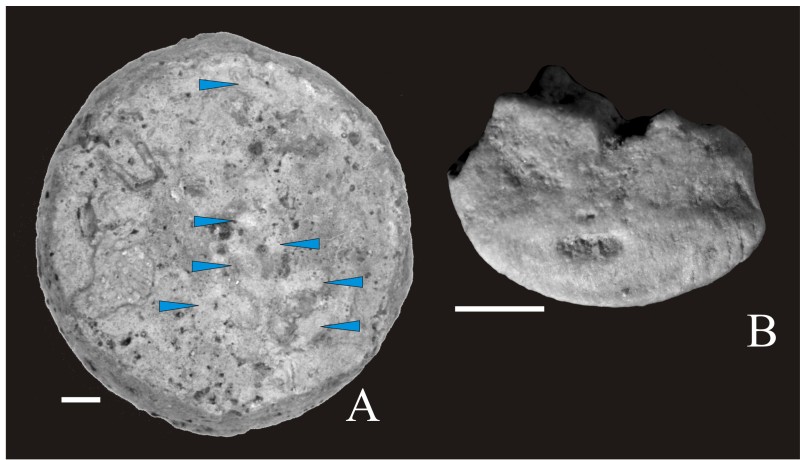

**Figure 3  Cyclocrinid remains from southern Poland.** (A) Columnal, articular facet. (B) Brachial plate. Scale bar equals one mm. Blue arrows mark granules covering the articular facet of columnal.

of two columnals and comes from the sample of Siedliska IG 1 (depth: 828.77 m). The other echinoderms observed on slab surfaces include numerous partly preserved cidaroid spines, cidaroid interambulacral plates and asteroid marginal plates. The echinoderm remains were also observed in thin sections and on the surface of polished slabs. In addition to these echinoderm remains, bivalves, gastropods and ostracods were observed both on the slab surfaces and in the macerated residues. In the late Bathonian sample Maciejowice IG 1, the remains of carbonized plant remains were also observed (for a summary of the above data see Table 4). In general, the Bathonian and Callovian crinoid assemblages are almost identical (see also Table 1). The other investigated Middle Jurassic crinoids were represented by numerous cups, cup remains, brachials, columnals, pluricolumnals, cirri, cirrals, and holdfasts (for details see *Głuchowski, 1987*; *Salamon & Gorzelak, 2007*; *Salamon & Zatoń, 2007*; *Salamon, 2008a*; *Salamon, 2008b*; *Salamon, 2008c*; *Salamon & Feldman-Olszewska, 2018*).

## State of preservation

The preservation of crinoids was examined only on the non-macerated surfaces of fresh core fragments. These were screened for signs of abrasion, bioerosion traces, chemical alteration of ossicle structure, evidence of epibionts and predation traces. Most of the crinoids display a high degree of disarticulation (almost 100%). However, they are all well preserved and do not show any traces of lateral surface abrasion. Excellent preservation and lack of abrasion are also noticed for columnal articular surfaces; the crenulae, petal floors, perilumens and lumens are complete. The two pluricolumnals also show no signs of abrasion. Additionally, the studied crinoid remains are very rarely bioeroded (bioerosion was only noticed in case of three columnals in the form of small and rounded holes). These may be ascribed to acrothoracican cirripedes, algae, fungi, polychaetes, sipunculans, or to sponge activities (for more details see *Salamon & Zatoń, 2006*; *Zatoń, Villier & Salamon, 2007*; *Salamon & Gorzelak, 2010*).

*Donovan (1991)* suggested that under normal oxygenated conditions, complete crinoid disarticulation takes place within two weeks. However, *Brett, Moffat & Taylor (1997)* argued that under certain circumstances, complete disarticulation may occur even after a year. In living comatulids, disarticulation starts just after death and by the end of the first week, only isolated calyx and a few arm fragments remain articulated (*Ausich, 2001*). Under high water temperatures and with increased physical disturbance, the disarticulation process significantly speeds up (see *Ausich, 2001*; *Hess, 2006*). *Baumiller & Ausich (1992)* noted that after 19 days, the isocrinid stem disarticulates into noditaxes and after 22 days, the column segments get disarticulated into isolated columnals. *Gorzelak & Salamon (2013)* noted that under high-energy conditions (constant transportation), and at room temperature, disarticulation of the crinoid skeleton is nearly complete after 17 days, where only a few articulated arms and cirral ossicles remain intact.

In the present study, the crinoid remains are preserved as cups, isolated columnals, brachials, and cups. Therefore, these remains can confidently be classified as taphonomic types 2 and 3 *sensu Brett, Moffat & Taylor (1997)*. Type 2 includes isocrinids with some remains of cyrtocrinid that undergo rapid postmortem disarticulation, on account of weak sutures. Type 3 comprises of cyrtocrinid cups in which major portions of the skeleton are resistant to disarticulation.

Thus, for the present study, all the data suggest that the recorded crinoids are autochthonous with negligible or no post-mortem transportation. Possibly, after death, they remained for some time on the sea bottom, but were not subjected to significant transportation or reworking before burial, or they were shortly covered by sediment, as suggested by the lack of abrasion surfaces. Bearing this in mind we assume little time-averaging that might alter the palaeoecological and palaeodepth information. Additionally, no trace of chemical alteration, predation traces and epibionts were noticed, suggesting that the inferred palaeoecological and palaeodepth signals are primary.

## Statistical analyses
### Substrate dependency (Hypothesis 1)
The ANOVA test (Table 5a) yielded statistically significant differences among the fauna of the different substrate types (claystones, sandstones and limestones) at $p < .001$. For pairwise comparisons, the Tukey's HSD (honestly significant difference) test was applied (Table 5b). This yielded a significant difference between both claystones-sandstones and limestones but no significance difference between claystones and sandstones (both being siliciclastics; Table 5b).

### Palaeodepth-dependency (Hypothesis 2)
The comparison between Palaeodepth parameters (near shore, shallow-marine, mid-ramp and offshore) did not yield any statistically significant results (Table 5c). The pairwise comparisons, Tukey's HSD test (Table 5d) retained a statistically significant difference only between the first two variables (Near shore and Shallow marine); although, the $p$ value is very low ($p > 0.01$).

**Table 5** **Analysis of variance (ANOVA).** (A) ANOVA test between different substrate types (Claystones, Sandstones and Limestones). (B) Tukey's HSD (honestly significant difference) test for pairwise comparisons. (C) ANOVA test between different palaeodepth parameters (Near shore, Shallow-marine, Mid-ramp and Offshore). (D) Tukey's HSD (honestly significant difference) test for pairwise comparisons.

**a**

| Source | SS | df | MS | |
|---|---|---|---|---|
| Between-treatments | 96 | 2 | 48 | $F = 8.57872$ |
| Within-treatments | 335.7143 | 60 | 5.5952 | |
| Total | 431.7143 | 62 | | |

**b**

| Pairwise comparisons | | | $HSD_{.05} = 1.7543$ <br> $HSD_{.01} = 2.2104$ | $Q_{.05} = 3.3987$ <br> $Q_{.01} = 4.2822$ |
|---|---|---|---|---|
| Claystones: Sandstones | $M1 = 0.90$ <br> $M2 = 0.33$ | | 0.57 | $Q = 1.11$ ($p = .71491$) |
| Claystones: Limestones | $M1 = 0.90$ <br> $M3 = 3.19$ | | 2.29 | $Q = 4.43$ ($p = .00748$) |
| Sandstones: Limestones | $M2 = 0.33$ <br> $M3 = 3.19$ | | 2.86 | $Q = 5.54$ ($p = .00068$) |

**c**

| Source of variation | SS | df | MS | |
|---|---|---|---|---|
| Between-treatments | 28.9 | 3 | 9.6333 | $F = 3.32335$ |
| Within-treatments | 220.3 | 76 | 2.8987 | |
| Total | 249.2 | 79 | | |

**d**

| Pairwise Comparisons | | | $HSD_{.05} = 1.4143$ <br> $HSD_{.01} = 1.7333$ | $Q_{.05} = 3.7149$ $Q_{.01} = 4.5530$ |
|---|---|---|---|---|
| $T_1:T_2$ | $M_1 = 0.35$ <br> $M_2 = 2.00$ | | 1.65 | $Q = 4.33$ ($p = .01563$) |
| $T_1:T_3$ | $M_1 = 0.35$ <br> $M_3 = 0.85$ | | 0.5 | $Q = 1.31$ ($p = .78958$) |
| $T_1:T_4$ | $M_1 = 0.35$ <br> $M_4 = 1.20$ | | 0.85 | $Q = 2.23$ ($p = .39675$) |
| $T_2:T_3$ | $M_2 = 2.00$ <br> $M_3 = 0.85$ | | 1.15 | $Q = 3.02$ ($p = .15112$) |
| $T_2:T_4$ | $M_2 = 2.00$ <br> $M_4 = 1.20$ | | 0.8 | $Q = 2.10$ ($p = .45097$) |
| $T_3:T_4$ | $M_3 = 0.85$ <br> $M_4 = 1.20$ | | 0.35 | $Q = 0.92$ ($p = .91521$) |

## DISCUSSION

### Depositional environment of the eastern part of the epicontinental Polish Basin

Upper Bathonian and Callovian of the eastern part of Polish Basin are represented by organodetritic, mostly crinoidal limestones with abundant limonite. There are grainstones and packstones which are dominated by echinoderm plates, less frequently appear echinoid spines, bryozoans, bivalves and brachiopods, occasionally gastropods, foraminifera, ammonites and belemnites are present. Quartz grains are frequently an admixture. In places also ferruginous ooids are present. Sedimentary environment is interpreted as a mid-ramp located between normal and storm wave base (*Feldman-Olszewska, 2018*).

### Depositional environment of the Łuków glacial draft

The Łuków glacial drift exposed in eastern Poland ('C' in Fig. 1), is a 30 m-thick Callovian unit containing black clays with carbonate concretions (see also *Mizerski & Szamałek, 1985*), deposited in an offshore setting (*e.g., Olempska & Błaszyk, 2001; Kaim, 2004*). However, recently, based on the presence of a high diversity asteroid assemblage, a varied environment (ranging from coastal to deep-shelf) was proposed (*Villier, 2008*). This unusually high crinoid diversity is likely a reflection of ossicles being transported by currents from shallow- to deep-water offshore environments (*Villier, 2008*). However, according to *Salamon (2008a)*, based on the ecology of crinoids, the Łuków clays was deposited in a relatively shallow environment and contrary to asteroids, they were not transported. Based on the presence of macro- and micro-remains of land flora, a nearshore environment was also inferred (see also *Brand, 1986; Marynowski et al., 2008*). Similarly *Gedl (2008)* using dinoflagellate cysts concluded that these originated presumably from shallow marine environments. Sedimentological investigations of the Aalenian-Bathonian (Middle Jurassic) claystones and shales from central Poland indicate that their sedimentation took place in offshore (below storm wave base) and transition (between normal and storm wave base) zones of the epeiric sea (see *Feldman-Olszewska, 2007; Feldman-Olszewska, 2008; Feldman-Olszewska, 2012a; Feldman-Olszewska, 2012b; Pieńkowski et al., 2008*). Water depth corresponds to the *Cruziana* ichnofacies but shallower than the *Zoophycus* ichnofacies (see *Knaust & Bromley, 2012*). The same depositional environment is expected for the Callovian clays from Łuków.

### Problems with isolated isocrinid classification

The isocrinids are a dominant crinoid group in the Middle Jurassic sediments of Poland. However, their correct classification into families and genera is very complicated. The ideal situation is when complete crinoid findings are available (*e.g., Simms, 1989; Hess & Messing, 2011*), but these are rare due to the unique structure of their skeletons (see *State of preservation*). There are of course also isocrinids that have a unique structure of columnals and it is easier to identify them from others. A classic example is that of *Balanocrinus subteres* or *B. pentagonalis* (see *Klikushin, 1992; Hess, 2014a; Hess, 2014b; Krajewski, Olchowy & Felisiak, 2016; Krajewski, Olchowy & Salamon, 2019; Krajewski, Ferré & Salamon, 2020*). *Hess (2014a)* and *Hess (2014b)* stated that the columnals of *B. subteres*

could be associated with its isolated calyx elements. Likewise, with remains of *Pentacrinites dargniesi* or *Isocrinus nicoleti* display a unique shape and ornamentation of their articular facets, as also *P. dargniesi* with its characteristic, ellipsoidal cirrals.

However, there is an issue with non-balanocrinid isocrinids, that have columnals of a similar morphology, and were erected on the basis of isolated skeletal remains. There are many such forms in the Middle and Upper Jurassic strata of Poland. According to *De Loriol (1877-1879)*, the most similar taxon to *Isocrinus amblyscalaris* is *I. munieri*. But there are some noticeable differences between them, also *i.e.,* the height of columnals and the ornamentation of the latera. *Hess (2014a)* added that *Pentacrinus pellati* with low columnals alternating with higher ones, and ridges occurring on its latera, is also a very similar form. He concluded that both species are of middle Oxfordian age and therefore may be conspecific. In turn, *Bather (1898)* suggested that *I. amblyscalaris* currently from the Oxfordian and Kimmeridgian of Poland is a synonym of *I. pendulus*. *Hess (1975)* added that both these species originated from the same middle Oxfordian strata (Bärschwil Fm, NW Switzerland). On the other hand, *Klikushin (1992)* stated that there is a series of morphological differences between these two species that supersedes *Bather*'s (*1898*) opinion. The latter author, elsewhere in his monograph, however, proposed to consider *I. pendulus*, Oxfordian *I. amblyscalaris*, and Callovian-Oxfordian *I. oxyscalaris* as junior synonyms of the Oxfordian *I. desori*. *Krajewski, Olchowy & Salamon (2019)* differed from this view and emphasized that there are some differences in the facet morphology of the mentioned taxa. Apart from the differences in the articular facets (*i.e.,* the presence of deep furrows between petal floors), the shape of columnals may also be different (compare *Hess, 1975*, p. 55, text-fig. 8, pl. 19, fig. 4; *Klikushin, 1992*, pl. 11, fig. 1–7 *vs.* pl. 13, figs. 8–10; *Radwańska, 2005*, fig. 8/4; *Salamon, 2009*, fig. 2H).

Taking into account all of the above, it is difficult to say unequivocally whether there are one or more biological species within the Middle/Upper Jurassic strata of Poland. Therefore, such forms, whose taxonomic affiliation raises serious doubts (*Balanocrinus berchteni, Isocrinus bajocensis, I. bathonicus,* and *I. pendulus*), must be evaluated with great caution in any analysis.

## Taxonomic status of cyclocrinids

The cyclocrinids (Cyclocrinidae) were classified by *Biese (1935-37)* as millericrinids (Millericrinida, Apiocrinitidae). Sieverts-Doreck (*Ubaghs, 1953*) and later *Hess (1975)* placed the cyclocrinids within cyrtocrinids (Cyrtocrinida). According to *Radwańska & Radwański (2003)* cyclocrinid columnals are elements of radical cirrals and therefore belong to bourgueticrinids (Bourgueticrinida). Following *Hess (2008)*, the ordinal position of *Cyclocrinus* and of the family Cyclocrinidae is left in the open nomenclature though the form has some resemblance to the Early Jurassic millericrinid genus *Amaltheocrinus*. *Hess & Messing (2011)* placed *Cyclocrinus* within uncertain order pending proper classification of its cup plates that still remain unknown.

One of the previously investigated Callovian sections (the abandoned Wysoka Quarry placed within the Polish Jura Chain; 'A' in Fig. 1), exposes sandy limestones where numerous cyclocrinid columnals were collected from a narrow level. Remains of other

crinoids were not found there. Along with these columnals, several "millericrinid"-like brachial (see Fig. 3) plates were found, which probably belong to the same crinoid as the columnals. Brachials display a proximal synostosis and a muscular distal side, sometimes with a pinnule socket. Both facets are parallel. This seems to confirm *Hess*'s (*2008*) idea of linking cyclocrinids to millericrinids. On the other hand, the "bourgueticrinid"-like, strongly abraded cup was found by us in the Czerwieniec locality by one of the authors (BJP). This cup is very similar in shape and size to bourgueticrinid cup that may suggest that cyclocrinids really belong to bourgueticrinids as suggested by *Radwańska & Radwański (2003)*.

## Cyclocrinids from Poland

*Wójcik (1910)* first listed the occurrence of *Cyclocrinus macrocephalus* in Poland and the occurrence of columnals in the sands of Filipowice, Polish Jura Chain ('A' in Fig. 1). *Głuchowski (1987)* noted the occurrence of *Cyclocrinus rugosus* and *Cyclocrinus* sp. within the Bathonian and/or Callovian crinoidal limestones of the Tatra Mountains and the Pieniny Klippen Belt ('E' and 'F' in Fig. 1). *Radwańska & Radwański (2003)* questioned this assignment, claiming that the sketches and photographs are not of sufficient quality, specimens are poorly preserved, and that the 'cyclocrind' were small-sized. Re-examinations of *Głuchowski* specimens (1987, fig. 13/1, 13/4 and pl. 1, fig. 1–6), stored at the Institute of Earth Sciences of the University of Silesia in Katowice, Poland, prove that they are significantly smaller than the columnals collected from other areas. Additionally, they are cylindrical, low and with a slightly convex latera, suggesting that they may well belong to cyclocrinids. Therefore, the original designations of *Głuchowski (1987)* are currently upheld. Moreover, *Radwańska & Radwański (2003)* recognized four cyclocrinid species within the Middle and Upper Jurassic strata of Poland, namely: *Cyclocrinus rugosus* from the Bajocian, *C. macrocephalus* from the Callovian, *C. areolatus* from the Oxfordian, and *C. couiavianus* from the upper Oxfordian. They stated that all mentioned taxa (with the exception of *C. couiavianus*) lie within the variability of *C. rugosus* (*Radwańska & Radwański, 2003*). Though describing columnals with articular surfaces covered by numerous and irregular tubercles, *Salamon & Zatoń (2007)* included these remains under *C. macrocephalus*. Re-observation of columnals coming from the same Callovian levels of the Polish Jura Chain, both ornamented by and devoid of tubercles, confirm the suggestion of *Radwańska & Radwański (2003)* that all these cyclocrinids actually belong to a single species, *Cyclocrinus rugosus*.

## Distribution of crinoids within particular facies. Does it really work?

Hans Hess in several of his papers pointed out that the occurrence of certain species of crinoids may be closely related to or strictly defined by substrate (sediment type) and palaeodepth. He claimed that *Chariocrinus andreae* inhabited muddy bottoms and its dense colonies occurred in somewhat deeper waters, below fair-weather wave base, *i.e.,* at depths around 10–15 m (*Hess, 1972*; *Hess, 1999a*). In this study, *Chariocrinus andreae* occurs in large amount in both carbonate and siliciclastic rocks. *Balanocrinus subteres* is one of the most cosmopolitan crinoid ever recorded in the Mesozoic and is known from

both shallow and deeper environments (compare Table 3 and Fig. 2). *Hess (1999a)* claimed that another very common Middle Jurassic crinoid taxon, *Isocrinus nicoleti*, frequently occurring in Poland, may also occur within both ooid shoals of the carbonate shelf and in slightly deeper environments (intercalating marls and bioclastic limestones). *Tang, Bottjer & Simms (2000)* recorded *I. nicoleti* from the shallow-water tidal facies of the Middle Jurassic Carmel Formation in Mount Carmel Junction and also from the open shallow subtidal (carbonate) facies of the Gypsum Spring and Twin Creek Formations of Wyoming (USA). *Hunter & Underwood (2009)* further noticed that *I. nicoleti* from the Bathonian of England and France, occurred only in shallow-water sediments, represented by low-energy, inner muddy shelf and outer lagoon, oolitic high-energy shoal system, carbonate/muddy low-energy, marine lagoon complex and in the shallow subtidal zone. All these are actually consistent with current observations that indicates *I. nicoleti* (Tables 2 and 3) occurred in both carbonate and siliciclastic rocks, although it preferred shallow-water carbonates. *Hunter & Underwood (2009)* concluded that the distribution of crinoids corresponds quite nicely to particular facies (but see also the comment on this paper by *Salamon, Gorzelak & Zatoń (2010)* and reply to it by *Hunter & Underwood (2010)*). This substrate-dependence is also corroborated by the ANOVA test (although at the generic-level) that the crinoid distribution is highly sensitive for carbonates (limestones) as opposed to siliciclastics (Table 5a–5b).

Of interest is the observation on *Pentacrinites dargniesi*. *Hunter & Underwood (2009)* also collected *P.* cf. *dargniesi* from shallow, muddy and carbonate habitats. *P. dargniesi*, despite being predominantly found in carbonates, also occurs in offshore claystones of Poland (see Tables 2 and 3). Contextually, it must be keep in mind that *P. dargniesi* was a pseudo-planktonic crinoid that transported on the lower surfaces of driftwood fragments, and hence, is likely to occur in varied facies (see *Simms, 1999*; *Hess, 1999a*).

The most numerous isocrinids within the Middle Jurassic of Poland are balanocrinids (*Balanocrinus subteres* and *B. pentagonalis*). *Hess (2014a)* and *Hess (2014b)* argued that balanocrinids settled mostly on hardgrounds but they are also be found these in mudstone-dominated settings, possibly on swells of the fine-grained bottom. The largest species of the slender crinoid genus (viz., *B. subteres*) was reported from below storm wave-base (*Hess & Spichiger, 2001*), where it thrived due to ample nutrient availability and well-oxygenated bottom waters (*Hess, 2014a*; *Hess, 2014b*). *Klikushin*'s (*1992*) noted, as also in the present study, that *Balanocrrinus subteres* is another most cosmopolitan Mesozoic crinoid species. It occurs in both carbonate and siliciclastic facies and in environments of varying depth, from near shore (close to a beach) to the open shelf, but usually below normal wave base (Table 3 and Fig. 2). This lack of palaeodepth-dependency (although at the generic-level) is also reflected in the ANOVA test that suggests that palaeodepth played no or minor role in the distribution of crinoids (Tables 5c–5d), although very weak (low $p$ value) significance is noted for near shore and shallow marine settings, as discussed above, also (see Fig. 2 and Table 5).

On the other hand, *Hunter & Underwood (2009)* concluded that the occurrences of balanocrinids (*Balanocrinus* cf. *subteres*) were restricted to neritic mudstone facies and brachiopod-rich limestones (*i.e.,* within the offshore, low-energy shelf setting). *Hunter &*

*Underwood (2009)* also noted that the comatulids show a wide range of distribution from offshore, low-energy shelf up to oolitic high-energy shoal system. In the present study, corroborating previous observations (*e.g.*, (*Hunter & Underwood, 2009*) (see also Table 5c–5d), the comatulids are restricted to siliciclastic rocks of near shore, below wave-base settings (Table 3 and Fig. 2).

The cyrtocrinids are considered as suggestive of relatively deep-sea depths and with carbonate facies association (*e.g.*, *Arendt, 1974*; *Hess, 1975*; *Žitt, 1973*; *Žitt, 1974*; *Žitt, 1975*; *Žitt, 1978a*; *Žitt, 1978b*; *Žitt, 1978c*; *Žitt, 1978d*; *Žitt, 1979a*; *Žitt, 1979b*; *Žitt, 1982*; *Žitt, 1983*; *Hess, Salamon & Gorzelak, 2011*). According to *Ausich et al. (1999)* and references cited therein), the cyrtocrinids prefer depths exceeding 100 m. Recent cyrtocrinids such as *Cyathidium foresti* live at depths ranging from 380 to 900 m, and *C. plantei* lives at a depth of 200 m (*Hess, 1999b*). Nevertheless, rare single reports also indicate their presence in shallow (and extremely shallow) environments (*e.g.*, *Salamon & Gorzelak, 2007*; *Salamon, 2019*; *Krajewski, Ferré & Salamon, 2020*). From the Jurassic Yátova Formation of Spain, *Zamora et al. (2018)* illustrated the spongiolithic facies as having been deposited in relatively shallow and open platform areas with depths not exceeding 60 m, containing the following cyrtocrinid cups: *Eugeniacrinites* sp., *Pilocrinus* sp., *Gammarocrinites* sp.; and a basal circlet of Tetracrinidae indet. that actually belongs to *Tetracrinus moniliformis* (see fig. 9e in *Zamora et al., 2018*). In the current study, *Gammarocrinites* sp., *Pilocrinus moussoni* and *Tetracrinus moniliformis* occur in carbonates of near shore and shallow marine, to deeper sublittoral under storm influence to open shelf environments.

The roveacrinids represented by the genus *Saccocoma* are exclusive to the shallow marine Bathonian and/or Callovian carbonates of the Tatra Mountains and the Pieniny Klippen Belt (Tethyan province). *Kowal-Kasprzyk, Krajewski & Gedl (2020)* mentioned saccocomids from the Oxfordian-Kimmeridgian exotic clasts of the Outer Carpathians in southern Poland. *Matyszkiewicz (1996)* and *Matyszkiewicz (1997)* mentioned *Saccocoma*-calciturbidites resting on the slope beds of Oxfordian cyanobacterial-sponge carbonate buildups formed in the Polish epicontinental basin (for summary see Fig. 2).

The cyclocrinids are not considered in the present study due to their controversial taxonomic assignment (*e.g.*, *Hess, 2008*).

Thus, qualitatively (see Fig. 2) isocrinids and cyclocrinids occur in both carbonate and siliciclastic rocks. The cyrtocrinids and roveacrinids occur within carbonate rocks, whereas the comatulids are exclusive to siliciclastics. In terms of palaeodepth, most crinoid groups dominate in shallow environments with the sole exception of cyrtocrinids, that are ubiquitous and occur in both shallow (near shore and shallow marine) and slightly deeper (deeper sublittoral to open shelf) settings. The occurrences of the cosmopolitan taxa, *Chariocrinus andreae* and *Balanocrinus subteres* (isocrinids), is independent of both substrate type and palaeodepth. This qualitative inference (Fig. 2) is largely mirrored quantitatively, also (Table 5).

The quantitative analysis (Fig. 2 and Table 5) suggests that the distribution of crinoids show a strong substrate-dependency (Hypothesis 1) (see Fig. 2 and Table 5) corroborating results by *Simms (1989)* and *Hunter & Underwood (2009)*. The palaeodepth-dependency (Hypothesis 2) did not yield significant results, although very weak (low *p* value) significance

is noted for near shore and shallow marine settings (Table 5), as also reflected in qualitative analysis (see Fig. 2). Although it must be accepted that the quantitative analysis (Table 5) is genus-based. Nevertheless, and until more species-based datasets are available, present quantitative analysis (Table 5) should prove useful in better understanding crinoid diversity trends. These results have important bearing on crinoid diversity studies and underline the importance of incorporating substrate details (facies analysis) while inferring distribution patterns of crinoid species diversity and palaeobiogeography. A more rigorous and quantitative species-based analysis may enable to better understand their varied distribution patterns.

## CONCLUSION

Almost all Bathonian and Callovian crinoids recorded from Poland dominate in relatively shallow environments. The only exception are cyrtocrinids, that are equally common in both shallow and deeper settings. The most commonly recorded Isocrinids, occur frequently in both carbonate and siliciclastic rocks. Cyrtocrinids also occur likewise, but dominate in carbonates. Roveacrinids are exclusive to carbonates, whereas cyclocrinids occur in both carbonates and siliciclastics. The comatulids occur exclusively in siliciclastics. The most cosmopolitan crinoids are the isocrinids, *Chariocrinus andreae* and *Balanocrinus subtrees*, that occur both in carbonate and siliciclastic rocks, and in shallow and open-shelf environments. Statistical analyses (ANOVA) corroborate these qualitative inferences, and suggests that the distribution of crinoids show strong substrate-dependency (Hypothesis 1) but not palaeodepth-dependency (Hypothesis 2), although statistical significance (albeit very weak; low $p$ value) is noted for near shore and shallow marine settings. A more exhaustive species-based analysis may enable to better understand their varied distribution patterns.

## ACKNOWLEDGEMENTS

Many friends, colleagues and students from the University of Silesia in Katowice (Tomasz Brachaniec, Przemysław Gorzelak, Jakub Jakubski, Rafał Lach, Mateusz Syncerz, Michał Zatoń, and some more whose names we do not remember) had been contributing through help during fieldworks and donating some crinoid specimens in years 2006-2020. Our reviewers (Aaron Hunter, Pedro Monarrez and an anonymous one) are acknowledged for critically reviewing the manuscript and providing constructive comments that considerably improved it. We also would like to thank Tomasz Wrzołek (University of Silesia in Katowice) for preparing polished slabs involved in the current study. We are also grateful to Alfred Uchman (Jagiellonian University) and Michał Zatoń for sharing references and providing subsequent literature. SJ thanks Ahmed Awad Abdelhady (Egypt) for confirming statistical results. Last but not least thanks go to Marcin Krajewski (AGH University of Science and Technology) for useful comments on the early draft of this MS.

### Funding

This work was not supported by any agency.

### Competing Interests

Bartosz Płachno is an Academic Editor for PeerJ.

### Author Contributions

- Mariusz A. Salamon conceived and designed the experiments, performed the experiments, analyzed the data, prepared figures and/or tables, authored or reviewed drafts of the paper, and approved the final draft.
- Anna Feldman-Olszewska conceived and designed the experiments, analyzed the data, prepared figures and/or tables, authored or reviewed drafts of the paper, and approved the final draft.
- Sreepat Jain performed the experiments, analyzed the data, prepared figures and/or tables, authored or reviewed drafts of the paper, and approved the final draft.
- Bruno B.M. Ferré and Bartosz J. Płachno performed the experiments, analyzed the data, authored or reviewed drafts of the paper, and approved the final draft.
- Karolina Paszcza performed the experiments, prepared figures and/or tables, and approved the final draft.

### Field Study Permissions

The following information was supplied relating to field study approvals (i.e., approving body and any reference numbers):

No consent was required as the samples were obtained in natural exposures or in inactive quarries. No consent is needed for this in Poland. There is no institution that could issue it.

### Data Availability

All examined specimens are housed in the Institute of Earth Sciences of the University of Silesia in Katowice, Poland, and catalogued under registration numbers: GIUS 8–2510, 8–2569, 8–2571, 8 - 3460Cr, 8-3466, 8-3678/1-6, 8-3734.

### Supplemental Information

Supplemental information for this article can be found online at http://dx.doi.org/10.7717/peerj.12017#supplemental-information.

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
