# Peer review of "Substrate type and palaeodepth do not affect the Middle Jurassic taxonomic diversity of crinoids"

_PeerJ, doi:10.7717/peerj.12017_

## Round 0.1 · original submission · Major Revisions

Thank you for submitting your manuscript " Does reservoir substrate type and palaeo-depth affect the Middle Jurassic taxonomic diversity of crinoids?" to PeerJ. I have now received reports from three reviewers and, after careful consideration, I have decided to invite a major revision of the manuscript.

As you will see from the reports copied below, the reviewers raise important concerns. I find that these concerns limit the strength of the study, especially Reviewer 2 thinks that the discussion does not link very well with the core data or the conclusions. Therefore I ask you to address all these issues with additional work. Without substantial revisions, I will be unlikely to send the manuscript back to review.

If you feel that you are able to comprehensively address the reviewers’ concerns, please provide a point-by-point response to these comments along with your revision. If you are unable to address specific reviewer requests or find any points invalid, please explain why in the point-by-point response.

·

Basic reporting

The hypotheses presented are sound, however, there is no research question/problem that is stated in the introduction to indicate the need to test these hypotheses. It is not until the discussion section that the problem is mentioned (i.e. the hypotheses posited by Hans Hess). The introduction can be greatly strengthened by explicitly stating the problem that the hypotheses would address. Additionally, lines 70–72 suggest that there might be distinct crinoid provinces in addition to those observed in ammonites. However, there is no mention of possible crinoid diversity differences between provinces in the results or discussion. Perhaps adding if/how crinoid diversity, paleodepth, and substrate occurrence differ between the two provinces would strengthen the overall effectiveness of the paper.

The use of “reservoir” in the context of this study is problematic. Reservoirs in a geologic context refer to accumulations of fluids (such as hydrocarbons and groundwater) within rocks. I believe “reservoir“ is used here to refer to depositional system (i.e. carbonate vs. siliciclastic dominated). I suggest avoiding the use of “reservoir” in this context.

The English language could be improved throughout the manuscript so that readers can clearly understand the information conveyed. I have annotated the manuscript in areas where there are grammatical mistakes with suggestions to improve for clarity.

Experimental design

Looking over the supplemental material, I see that comprehensive lithological data are missing. I don’t question the interpretation of water depth the authors have made as their results are consistent with other studies, but are there any other lithological data to strengthen the interpretations?

Validity of the findings

Conclusion section can be strengthened by linking results back to the research problem. The results of study suggest that some crinoids can be used to infer water depth while others can not. Additionally, there is no explicit summary for how water depth and substrate affect crinoid diversity as stated in the title. Furthermore, the discussion section could be strengthened by linking results back to the research problem more directly as well. Perhaps a final paragraph that summarizes whether crinoids can indeed be used to infer water depth and how water depth and substrate affect crinoid diversity.

Additional comments

Overall, this study is an intriguing test of whether crinoids can be reliably used to infer substrate type and water depth using the fossil occurrences from the Middle Jurassic of Poland. The authors are to be commended for their rigorous taxonomic approach as well as diligently presenting background information, methodology, and discussion to support their conclusions. Additionally, as far as I can tell, their taxonomic descriptions are sound. With that being said, this manuscript suffers from a few structural and grammatical issues that weaken the presentation of the study.

An option to consider to strengthen the paper further (but not required) would be to use quantitative methods to test whether various crinoids indeed have a habitat preference (i.e. water depth, substrate). If there are comprehensive lithologic data, they can be combined with ordination techniques to graphically demonstrate the association of crinoids and water depth and substrate. Ordination techniques such as non-metric multidimensional scaling or detrended correspondence analysis could be used, which are relatively easy to use. This approach could also be used to test whether there is evidence for crinoid provinciality within the Central European Basin. In case the authors are not familiar with these techniques, they find more information on ordination techniques, particularly, non-metric multidimensional scaling here: https://www.palass.org/publications/newsletter/palaeomath-101/palaeomath-part-13-mds-and-ordination and here: https://strata.uga.edu/software/pdf/mdsTutorial.pdf

·

Basic reporting

This MS is clearly written with good English. It appears rushed in places. Some English is informal. For example, they site Hunt & Underwood (2009) when it is Hunter & Underwood (2009). Hunt is distinct from Hunter as a family name. Balanocrinus subtrees does not exist maybe its Balanocrinus subteres? In addition, lots of key references are missing.

The MS is well cited throughout. The title suggested a broad study, However there is a bias towards the Polish literature and to the Middle Jurassic.

The structure of the confusing in its current state, the title and abstract suggest a study with a wide general significance. However, the introduction and data analysis is very specific and the discussion reads more like a literature review rather than focusing on the very useful data-set.

The data from the cores is certainly interesting, but the MS appears to be a mixture of limited new data from cores and a literature

It is not clear how the new data is relevant to the extensive discussions of previous research on Middle Jurassic. The figures and tables are limited for example Figure 2 “Model showing distribution of selected Callovian crinoids.” Why selected? Does this mean data has been omitted?

Experimental design

Part of this MS is “Original primary research within Aims and Scope of the journal” however, as parts of the data is limited is relies heavily on a literature review of previous research.

The core premise of the MS is to cast doubt that crinoids are, with a few exceptions, benthonic organisms are generally considered as good indicators for determining environmental conditions. This is an important question in the study of crinoids. This is a useful case study, but I am not sure the quality and preservation of the data is enough to fulfil the two very bold questions. Do Callovian crinoids occur equally and frequently in both deep and shallow water environmental settings (palaeo-depth dependency)? and (b) Do Bathonian and Callovian crinoids occur equally and frequently in both carbonate and siliciclastic facies (substrate dependency)? It is not clear from the discussion if these questions are answered. The results of the paper get lost in a long discussion of other studies. The conclusions are very general for example “Almost all Bathonian and Callovian crinoids recorded from Poland dominate in relatively shallow environments”. This observation is certainly not new. “The only exception are cyrtocrinids, that are equally common in both shallow and deeper settings.” Again, this is well known. “Isocrinids, most commonly recorded, occur frequently in both Carbonate and siliciclastic rocks” Again very general statement and not a new idea. The Research question needs to be better defined there is no discussion on where previous studies have failed to answer these questions. In contrast the methods on core analysis are useful can could be expanded.

Validity of the findings

All underlying data has been been provided but it is not robust in that the facies are generalised. The discussion reads too much like a literature review rather than a detailed discussion of the data provided in the tables. As a result, the conclusions need to be revised.

I am a bit confused as the MS confidently, says that “isolated remains of isocrinids, consisting of columnals, pluricolumnals, cirrals, cup plates, and brachials, are classified at the specific level.” Yet authors then discuss the Problems with isolated isocrinid classification Most of this section does not take into consideration recent literature on Pentacrinites

I have big problems with the “Distribution of crinoids within particular facies. Does it really work?” In that it compares species from very different environments are paleobiogeographic provinces and in some ways is oversimplified and is over focused on the Middle Jurassic. For example, Balanocrinus subteres might be a cosmopolitan crinoid in Poland. But this might not be the case in the UK or North America.

Additional comments

In Conclusion this is an interesting study, that questions if crinoids as possible indicators of Paleo Depth. As the authors point out there is ample research that demonstrates a link between robust proxies for inferring changes in salinity and sedimentation rate and for inferring substrate type. I am supportive of this MS and I believe it should be published in some form after a major revision. However in its current state I fail to see how the current MS challenges this observation with limited new data and a discussion of previous published data.

1. The title is a bit misleading, it suggested a global study, when in-fact it’s a local cases study from (a) archival Bathonian-Callovian (Middle Jurassic) crinoid occurrences from Poland and (b) newer finds from five borehole data from eastern Poland.

2. It is from a limited timescale Bathonian-Callovian. Despite this it makes a bold conclusion that use of crinoids as reliable proxies for inferring reservoir depth (paleodepth) and substrate type, should be considered with caution. Yet make these conclusions there needs to be discussion of crinoids outside the Jurassic. Such as the extensive literature on the Cretaceous and Palaeogene.

3. The sedimentology is over-generalised into “near shore close to beach” or shore face carbonate ramp” or Offshore/open shelf. This is does not take into consideration local variations within these defined facies. Sadly this is one of the issues in using borehole data.

4. The title uses the word reservoir, which should only be used in its correct context within Petroleum geology. I think indicators of palaeodepth are more appropriate

5. One of the achievements of studies such as Hunter and Underwood (2009) was that it looked a diverse set of Formations grouping these into lithofacies and in Hunter and Underwood (2009) tapho-facies. These facies covered a limited time interval of the Bathonian. However this study lumps data from the whole Middle-Late Jurassic.

Reviewer 3 ·

Basic reporting

The manuscript is exhaustively referenced, with many recent, high-quality publications cited.

The regional geologic background is provided and explained clearly, as is the basis for identification of crinoid elements. The taxonomic status of the major crinoid groups encountered is discussed, with classification issues addressed.

Data are provided, both in the form of novel data collected from analysis of new borehole sampling and a literature review.

There are a number of minor grammatical and formatting errors throughout the manuscript. It would be best to go through the manuscript thoroughly after making any substantive revisions to "clean up" any wording issues. Some examples are provided below:

1. Line 19: the first sentence needs to be re-worded (perhaps "...benthonic organisms THAT are generally..."

2. Line 34: Balanocrinus subteres is mis-spelled ("subtrees").

3. Line 52: an extra parenthesis is present.

4. Lines 73-76: this paragraph needs to be divided into several sentences or have some modified punctuation in order to be grammatically correct.

5. Line 261: "high dynamic" should probably be "highly dynamic".

6. Line 293: "alsoi.e."

7. Lines 404, 406, 412: The Hunter and Underwood article is listed in the references as 2009, not 2010.

Experimental design

This contribution represents original primary research. It also serves as a novel compilation and synthesis of previously described crinoid occurrences.

The techniques used to collect new data are described in sufficient detail for replication.

The criteria used to identify and classify recovered crinoid remains are outlined, consistently applied, and appear systematically sound.

The knowledge gap filled by this research and its broader significance is outlined, although the scope of this study is somewhat under-stated. For example, the discussion of taxonomic standardization and the systematic issues relevant to some Jurassic crinoids--which comprise a large portion of the manuscript--are not mentioned in the beginning of the paper.

Validity of the findings

In general, conclusions are supported by all available data. Interpretations are directly linked to data.

There are only a few substantive issues that I wanted to raise:

1. On lines 240-241, it is stated that all paleo-depth interpretations should be taken as accurate, as all crinoid remains are autochthonous. However, on lines 255-257, it is stated that the high diversity of asteroid ossicles in the Lukow glacial drift reflects transportation of elements into deeper water. This makes it seem as though at least some of the crinoid elements may have been allochthonous as well.

2. It is not clear why the description and discussion of the Lukow glacial drift is included on lines 251-272. What does this add to the manuscript?

3. At the end of the abstract (lines 37-38), it is stated that the use of crinoid remains for paleoenvironmental interpretation should be considered with caution. However, the results of the study demonstrate a number of consistencies and corroborate a number of previously suggested paleoenvironmental preferences. Moreover, the need for caution is not really brought up significantly in the discussion section and is not mentioned in the conclusions section. The fact that this statement comprises the end of the abstract makes it seem as though the need for caution is the most important take-home message of the paper, but this does not appear to be the case.

4. This is less significant, but (1) I don't like the use of the word "reservoir" in the title and throughout the manuscript, as it is not clear how this word is being used; and (2) the word "Polish" or "Poland" should probably occur within the title, as this is not a global survey of Middle Jurassic crinoids.

Additional comments

This is a highly ambitious contribution, and I commend the authors on their work. In particular, I appreciate the detailed research on primarily disarticulated specimens and isolated ossicles, which are generally under-studied.

---

## Round 0.2 · Minor Revisions

Reviewer 2 (Aaron Hunter) would like to see a short discussion on the influence of transport and abrasion on the crinoids sampled and if there is any time averaging. Please address this issue in the revised manuscript.

·

Basic reporting

With the change of title and the introduction this MS is much improved. This is a very useful study and should be published. It means the apparent bias towards the Polish literature fits well with the introduction to the MS. The revisions mean that the hypothesis is now clear with the twin objectives clearly spelt out.

Experimental design

I have no further issues with the experimental design as limitations of using data from boreholes his clear.

Validity of the findings

I commend the authors for a very good revision that re-frames the MS in the Polish context but now makes the MS significant case study that can be built on using other examples of crinoids from the Middle Jurassic and as the authors have said, this can now be applied to other parts of the stratigraphy. Before publication I would like to see a short discussion on the influence of time averaging, abrasion and transportation on the samples. They also point out that Hunter and Underwood used quantitative analysis for inferring crinoid distribution patterns. But actually, the samples were exhaustively sampled so that ALL the ossicles from the beds sampled and therefore taking into account the bias from preservation and time averaging. I look forward to seeing this published!

Additional comments

They have included Salamon, M.A., Gorzelak, P., Zatoń, M., 2010. Comment on “Palaeoenvironmental control on distribution of crinoids in the Bathonian (Middle Jurassic) of England and France” by Aaron W. Hunter and Charlie J. Underwood. Acta Palaeontologica Polonica, 55, 172–173.....Can they also include my reply as its relevant to the preservation of the crinoids?

---

## Round 0.3 · accepted · Accept

Thanks for your revisions and reply. Only one issue needs to be checked. Paleodepth/palaeodepth should be uniform in the manuscript .